# Durability and Electrical Conductivity of Carbon Fiber Cloth/Ethylene Propylene Diene Monomer Rubber Composite for Active Deicing and Snow Melting

**DOI:** 10.3390/polym11122051

**Published:** 2019-12-10

**Authors:** Shuanye Han, Haibin Wei, Leilei Han, Qinglin Li

**Affiliations:** College of Transportation, Jilin University, Changchun 130025, China; hansy18@mails.jlu.edu.cn (S.H.); hanll18@mails.jlu.edu.cn (L.H.); liql1150142@163.com (Q.L.)

**Keywords:** thermal behavior, multifunctional composites, mechanical properties, electrothermal properties, active deicing and snow melting, temperature durability

## Abstract

To reduce the impact of road ice and snow disaster, it is necessary to adopt low energy consumption and efficient active deicing and snow melting methods. In this article, three functional components are combined into a conductive ethylene propylene diene monomer (EPDM) rubber composite material with good interface bonding. Among them, the mechanical and electrical properties of the composite material are enhanced by using carbon fiber cloth as a heating layer. EPDM rubber plays a mainly protective role. Further, aluminum silicate fiber cloth is used as a thermal insulation layer. The mechanical properties of EPDM rubber composites reinforced by carbon fiber cloth and the thermal behaviors of the composite material in high and low temperature environments were studied. The heat generation and heat transfer effect of the composite were analyzed by electrothermal tests. The results show that the conductive EPDM rubber composite material has good temperature durability, outstanding mechanical stability, and excellent heat production capacity. The feasibility of the material for road active deicing and snow melting is verified. It is a kind of electric heating deicing material with broad application prospects.

## 1. Introduction

In most parts of the world, road ice and snow problems occur in winter, and various methods are adopted for snow melting and deicing, but the results are unsatisfactory. Every year, a large number of traffic accidents are caused by snow and ice disasters. At present, artificial and mechanical salt spraying are the main measures to protect against road icing. Various salt chemicals such as NaCl, MgCl_2_ [1] and potassium acetate [2] are used on the road. The method of salt spraying can melt snow and ice at the beginning of use. However, most of the salts used contain chloride ions, which have serious corrosive effects on steel [3] and concrete [4,5] in road infrastructure. Simultaneously, salt chemicals have a serious impact on the ecological environment around the road and groundwater [6,7,8]. In view of the problems of chemical deicing, to achieve the purpose of rapid and effective road deicing and snow melting that will not affect the ecological environment around the road, scholars have developed active snow and ice removal technologies.

Existing active deicing and snow melting technologies mainly include the conductive asphalt mixture method, heat pipe method, geothermal energy deicing method, and the electric heating method. The conductive asphalt mixture method mainly adds conductive fillers such as steel fiber [9], steel slag [10], graphite and carbon fiber [11,12] to the asphalt mixture. In this way, the purpose of reducing the electric resistance of the asphalt mixture is achieved. Then, the conductive asphalt concrete is heated by electric current or infrared heating [13,14], thereby realizing active deicing of the road. However, the mechanical properties of asphalt mixtures decrease due to the addition of conductive fillers. In addition, the electrical resistance value of the asphalt mixture with conductive filler is greater than that of conductive filler, resulting in low thermal efficiency. A faster heating rate can only be achieved when the input power is relatively large. This method takes a long time to achieve road snow melting and deicing, resulting in serious energy waste and a high cost of use. Heat pipe technology realizes the active snow melting and deicing of the road by embedding pipes in the road structure and introducing a heat flow into the pipes. Chen et al. [15] conducted a snowmelt test after laying heat pipes on the airport runway. The results proved that the heat pipe technology can achieve snow melting and deicing, which provided certain support for the actual construction design. Liu et al. [16] studied the effects of various factors such as wind speed, heating power, and buried spacing on the melting of electric heating tubes. However, pipeline laying had a certain impact on the structural layer of the road. There were high requirements for heat flow control and road construction during use. Geothermal energy is a clean energy source. Han et al. [17,18] controlled the geothermal heat pump system and the thermoelectric system to study the use of geothermal heat to achieve snow melting and the deicing of roads. Due to the limitation of geothermal energy and the low efficiency of thermoelectric conversion, it was impossible to melt snow in real time. Because the above active deicing methods have some defects, they have not been widely used. Conductive composite materials have developed rapidly in recent years, so the application of conductive composite materials on roads is considered. Active road deicing and snow removal can be realized by power on.

In order to achieve efficient and fast road deicing and snow melting, the effective reduction of resistance value has been proven to be one of the key ways to improve electrical heating performance [19,20]. Carbon series materials have good electrical conductivity. Therefore, Chu et al. applied materials with excellent conductivity such as carbon nanotubes [21] and graphene-paper [22,23] to deicing applications. The effect of actual deicing is obvious, but materials such as carbon nanotubes and graphene are expensive. Currently, coverage of the large areas occupied by roads is not easy to achieve. In comparison, carbon fiber cloth is a material that has developed rapidly in recent years. The cost of carbon fiber cloth has been greatly reduced due to the improvement of carbon fiber preparation technology. Since carbon fiber cloth has the characteristics of low resistance and good durability, it has been studied in the preparation of electrodes [24] and in 3D printing heating materials [25]. Furthermore, the mechanical strength of carbon fiber cloth is high. It has been widely used in the mechanical properties of reinforced composite materials [26] and the reinforcement of bridges [27]. In view of the excellent mechanical and electrothermal properties of carbon fiber cloth, Lim et al. [28,29,30,31] added carbon fiber cloth directly to the concrete, and then heated concrete by electricity. The heat production rate of the carbon fiber cloth and the effect of ambient temperature on heat production efficiency were investigated. The results showed that the carbon fiber cloth can quickly heat up when the input power is small, and achieve the purpose of rapid deicing and snow melting. Previous studies proved that carbon fiber cloth has broad application prospects in road deicing and snow removal. However, the protection of carbon fiber cloth and the durability of carbon fiber cloth composite material were not considered.

In this paper, a conductive ethylene propylene diene monomer (EPDM) rubber composite material with a three-layer structure was prepared using carbon fiber cloth as heat producing layer and sandwiched between a heat conducting layer of EPDM rubber and heat insulating layer of aluminum silicate fiber cloth. The effects of carbon fiber cloth on the mechanical and electrical properties of EPDM rubber were investigated. Combined with the environment in which the composite material was used on the road, the thermal behaviors of the composite material under a high temperature 60 °C environment and the durability of the material after cycling at −20 °C and normal temperature were analyzed. The heat generation uniformity and heat production effect of the composite material were studied. The results showed that the conductive composite material based on carbon fiber cloth has the characteristics of high efficiency, reliability, durability, high mechanical strength and low cost in deicing and snow melting. It is a potential electric heating deicing material in many fields.

## 2. Materials and Methods

### 2.1. Materials

Carbon fiber cloth used in this study was purchased from Shanghai Yuezi Industrial Co., LTD (Shanghai, China). The carbon fiber cloth is woven from 12 K carbon fiber precursors, with a carbon content of more than 98%, a thickness of 0.167 mm and a density of 300 g/cm^3^. Because of the good electrical conductivity of carbon fiber cloth, it was used as an electrical heating layer. Ethylene propylene diene monomer (EPDM) rubber (Sanhe Great Wall Rubber Co., LTD., Sanhe, China) with a density of 0.90 g/cm^3^ was used as a protective and bonding material for carbon fiber cloth. To transfer heat rapidly, graphite (Tianjin Dengke Chemical Co., LTD., Tianjin, China) with high thermal conductivity was added to EPDM rubber. The carbon content of graphite is more than 99.93% and the particle diameter is 6.5 μm. The addition of carbon black with a particle size of 40 nm and auxiliary materials (Kaiyin Chemical Co., LTD., Kaiyin, China) such as vulcanizing agent dicumyl peroxide to EPDM rubber improve the mechanical properties and durability of the composite material. The obtained thermally conductive EPDM rubber composite material was used as a heat transfer layer. The heat insulation layer was composed of aluminum silicate fiber cloth (Dacheng North Sealing Products Factory, Dacheng, China) with a low thermal conductivity of 0.035 W/(m·K) and EPDM rubber. Finally, the three-layer structure was vulcanized at high temperature and high pressure to obtain a conductive EPDM rubber composite material.

### 2.2. Preparations of Conductive EPDM Rubber Composite Material

The conductive EPDM rubber composite material is composed of a heat transfer layer, a heat generating layer and a heat insulating layer from top to bottom. First, a thermally conductive rubber compound was prepared. EPDM rubber, graphite, carbon black and vulcanizing agent were mixed at a weight ratio of 10:4:4:0.2, and placed in an open mill (Qingdao Shunfu Rubber Machinery Manufacturing Co., LTD., Qingdao, China) at room temperature. Mixed for 30 min to distribute the fillers evenly in the rubber. To simplify the process, the obtained rubber compound was used as a binder for the carbon fiber cloth and the heat insulating layer aluminum silicate fiber cloth. Second, the mixture was left at room temperature for 30 min while the press vulcanizer was preheated and two kinds of fiber cloth and steel molds were prepared. Then, an aluminum silicate fiber cloth, a rubber compound, a carbon fiber cloth, and a rubber compound were placed in this order from bottom to top in the steel mold. The steel mold was placed in the press vulcanizing machine which provided a high temperature and high pressure environment of 170 °C and 10 MPa. Finally, the conductive EPDM rubber composite material was obtained by vulcanization for about 10 min. The density of conductive EPDM composites is 1.25 g/cm^3^. The electrothermal sample size was 1000 × 500 × 6.5 mm^3^. Mechanical test specimens were prepared according to the corresponding test requirements. The sample preparation process is shown in Figure 1.

### 2.3. Measurements

In order to determine the feasibility of conductive EPDM rubber composite material for road deicing and long-term use in roads, mechanical properties, and electrothermal properties of the composite material were measured.

#### 2.3.1. Mechanical Properties Tests

The influence of carbon fiber cloth on mechanical properties (tensile strength, tear strength and compressive strength) of EPDM rubber composite material was determined by an electronic universal testing machine (Metis Test Machine Factory, Tianjin, China). The effects of carbon fiber cloth on the mechanical properties of EPDM rubber composite material were determined by mechanical experiments. The mechanical properties of the composite were measured according to the methods [32,33,34] for measuring the mechanical properties of hot vulcanized rubber composite material. The tensile tests were performed using dumbbell-shaped tensile test specimens, and the test area of the sample was 20 × 5 × 6.5 mm^3^. The non-experimental zone of the sample was clamped in the chuck of the electronic universal testing machine, and the moving speed was adjusted to 50 mm/min. The sample and the post-test sample are shown in Figure 2. The tear tests were carried out using trouser samples, the moving speed of the chuck was 100 mm/min, and the size of the tear test sample was 100 ×15 × 6.5 mm^3^. The samples used for the compression tests were circular samples of Φ 29 × 6.5 mm. The samples were compressed at a speed of 10 mm/min. This test was cycled four times, and the result of the fourth compression test was recorded. The thickness of all the tested samples was 6.5 mm, and the force-displacement curves during the mechanical tests were obtained by the electronic universal testing machine.

#### 2.3.2. High Temperature Durability Tests

Since the conductive EPDM rubber composite material was used in roads, durability studies were conducted based on the most unfavorable conditions of road temperature. According to data from the national meteorological information center of China, the highest road temperature in summer is around 60 °C, and the average temperature in winter is around −20 °C in Northeast China. Temperature has a great impact on the durability and service life of the material. Therefore, the samples were placed in a blast oven (Sanshui Scientific Instrument Co., LTD., Tianjin, China). The high temperature aging tests were carried out at a high temperature of 60 °C for 24, 48, 72, 96, and 120 h. The changes of tensile strength, tear strength, and compressive strength of the material at different times were measured. Three parallel tests were performed for each test and test errors were indicated by error bars.

#### 2.3.3. Low Temperature Cycle Tests

The samples were placed in a refrigerator at −20 °C for low temperature cycle tests. After 24 h of freezing, the samples were taken out and thawed at room temperature of 26 °C for 24 h, and this was repeated 20 times. Part of the samples were taken out for mechanical tests every 5 cycles. The tensile, tear and compressive properties of the conductive EPDM rubber composite material after low temperature cyclic treatment were evaluated by the electronic universal testing machine.

#### 2.3.4. Heat Production Uniformity Tests

As shown in Figure 3b, copper electrodes (Figure 3a) with the size of 500 × 10 × 1.5 mm^3^ and the carbon fiber cloth are fixed using clips. The addition of the copper electrodes can effectively reduce the contact resistance and achieve a uniform distribution of voltage. In order to test the uniform heat production of carbon fiber cloth in circuit. As shown in Figure 3c, the temperature sensors (Changsha Sanzhi Electronic Technology Co., LTD., Changsha, China) 1, 2 and 3 were placed in the middle, the middle of the edge and the corner portion of the sample, respectively. The temperature changes of the three temperature sensors were measured. The connection of the circuit is shown in Figure 4. The power supply was converted from AC 220 V to low voltage by means of a transformer, and the sample size used in the tests was 1000 × 500 × 6.5 mm^3^. Electrothermal tests were carried out at room temperature without thermal insulation measures and the effects of wind. By inputting different power, the change of temperature with the time of electrification is recorded. Thereby, the uniformity of the conductive EPDM rubber composite material was determined.

#### 2.3.5. Resistance Stability and Thermal Efficiency Tests

Through electrothermal tests, the change of current and voltage in the circuit is measured. The resistance value of the composite material after multiple input of different power was calculated to analyze the resistance stability of the composite material. Temperature sensors were placed on the upper and lower layers of the material to measure the change rate of temperature after inputting different power. Thus, the thermal efficiency of the composite and the function of thermal insulation layer were analyzed.

## 3. Results and Discussion

### 3.1. Effect of Carbon Fiber Cloth on Mechanical Properties of EPDM Rubber Composite

Figure 5 combines two typical force-displacement curves of EPDM rubber composite materials with and without carbon fiber cloth in one coordinate system. As shown in Figure 5, the EPDM rubber composite materials have typically three stages of elasticity, yield, and strain hardening in the process of stretching. Further, the influence of carbon fiber cloth on the three stages is different. After the addition of the carbon fiber cloth, the corresponding maximum value of the composite material in the elastic phase increases. According to the formula:(1)$$TS=\frac{F_{m}}{Wd}$$
where *TS* is the tensile strength (MPa), *F*_m_ is the maximum force recorded (N), *W* is the width of the test area (mm), and d is the thickness of the test area (mm).

The *F*_m_ increases after the carbon fiber cloth is added, so the tensile strength is remarkably improved. However, carbon fiber cloth breaks when the tensile force reaches the maximum value, so the force falls quickly and it has less influence in subsequent stages. After that, the conductive EPDM rubber composite material shows a yielding phenomenon. The yield force of the composite material with carbon fiber cloth is slightly higher than the composite material without carbon fiber cloth. In the strain hardening stage, the carbon fiber cloth in the composite material breaks. Therefore, the composite material with carbon fiber cloth overlaps with the tensile curve of the composite material without carbon fiber cloth. According to Figure 6, the change trend of the two tear test curves is similar, but the maximum values of the tear process force are different. The tearing force value of the added carbon fiber cloth is larger than that of the composite material without the carbon fiber cloth. Because the tear sample size is the same, it can be concluded that the addition of carbon fiber cloth can increase the tear strength of the composite material. According to Table 1, it can be concluded that the addition of carbon fiber cloth increases the tensile strength of the composite by 78.3% and the tear strength by 23.1%, but the addition of carbon fiber cloth increases the compressive strength of the EPDM rubber composite by only 0.5%.

Since the mechanical strength of the carbon fiber cloth in the fiber direction is high, the tensile strength and the tear strength of the composite material are relatively large. However, the compression tests were perpendicular to the plane in which the carbon fiber cloth was laid, and the fibers were only subjected to the compressive force during the compression process. And the thickness of carbon fiber cloth is only 0.167 mm. As a result, the compressive strength of the composite material was mainly provided by the EPDM rubber. The carbon fiber cloth had substantially no effect on the compressive strength of the EPDM rubber composite material. In summary, the addition of carbon fiber cloth has greatly improved the tensile strength and tear strength of the conductive EPDM composite material. The application of the conductive EPDM rubber composite material on the road is guaranteed.

### 3.2. Durability of Conductive EPDM Rubber Composite Material at 60 °C

It can be clearly seen from Figure 7 that after the high temperature treatment, the maximum value of the force during the stretching process increases, thus the tensile strength of the composite material increases. However, the yield point of the composite treated at 48 h is lower than that of the composite material without high temperature treatment. The strain hardening stage is also lower than that of untreated samples. Figure 8 is the effect of high temperature duration on the tensile strength of the composite material. Curve fitting was performed on the data obtained from the tensile tests, and the exponential function was obtained by fitting:(2)$$y=-2.596e^{-\frac{x}{42.364}}+9.279$$
where *x* is the heating time (h) and *y* is the tensile strength (MPa). The correlation coefficient is 0.9858. According to the characteristics of the exponential function, the mechanical properties of the material under long-term high temperature environment can be predicted appropriately. Further, the tensile strength of the composite material tends to stabilize gradually. With respect to Table 2 and Figure 9, the tensile strength, tear strength and compressive strength of the composite material are significantly improved after high temperature treatment for 72 h, which increase by 38.04%, 8.78%, and 33.85%, respectively. The tensile strength, tear strength and compressive strength after 120 h high temperature treatment increase by 2.70%, 0.11% and 1.95%, respectively, compared with 72 h. The mechanical properties do not change substantially, and tend to be stable.

After 120 h of continuous high temperature treatment, the mechanical properties of the conductive EPDM rubber composite material have been improved. The reason why the composite material was greatly improved before 72 h was that no secondary high temperature vulcanization was performed during the preparation of the samples. The residual vulcanizing agent and other materials in the composite did not react adequately [35]. The reaction rate of the residual material was accelerated in a 60 °C environment, resulting in a rapid increase in mechanical properties. But the breaking strength of the composite material decreased. After 72 h, the residual material was substantially completely reacted. The subsequent high temperature treatment had little effect on the mechanical properties of conductive EPDM rubber composite material. Because the heating temperature is relatively low and the high temperature aging resistance of EPDM rubber is good [36], there is no phenomenon in which the performance of the rubber composite material is degraded. It is less likely that special high temperatures will occur on roads, and the duration will not be long. It is proven that conductive EPDM rubber composite material has good durability and can be used in road for a long time.

### 3.3. Mechanical Properties After Low Temperature Cycling

According to Figure 10, as the number of low temperature cycles increases, the tensile strength, tear strength, and compressive strength curves of the conductive EPDM rubber composite material change little, and there is no significant downward or upward trend. The test data in Table 3 show that the tensile strength, the tear strength and the compressive strength fluctuate around the average of 6.42 MPa, 16.51 KN/m and 3.70 MPa, respectively. Compared with the mean values, the fluctuation values are 0.47%, 0.61% and 3.78%, respectively. The range of curve fluctuations is between the test error bars of the parallel tests. The small fluctuations can prove that the mechanical properties of the composite material are relatively stable, and there is no large performance degradation phenomenon with multiple low temperature cycles. The test data and curve show that the composite material has good low-temperature durability, and the mechanical properties do not change greatly with the increase of the number of low temperature cycles times. This proves that the conductive EPDM rubber composite material has good low temperature durability and can be used for a long time in the road environment with a large temperature difference.

### 3.4. Heat Production Uniformity of Conductive EPDM Rubber Composite Material

Conductive EPDM rubber composite material is designed to quickly solve road ice and snow disasters. The heat generation uniformity, resistance stability, and heat generation effects of different input power need to be analyzed before the actual snow melting test. In this way, it is possible to ensure the safe and efficient operation of the conductive EPDM rubber composite material in the road.

Figure 11 shows the temperature change of the upper layer of the sample at different input power for the temperature sensors at three locations. Meanwhile, a control temperature sensor was placed at room temperature to determine the variation of room temperature. Due to instrumental errors in the temperature sensors, there are differences in the initial temperatures of the three temperature sensors. As can be seen in Figure 11, the three temperature sensors have the same rising speed, and the three curves are parallel. According to Table 4, after 30 min of energization, the three temperature sensors show that the temperature changes in the upper layer of the sample are consistent. There is no change in room temperature, so the rise of the sample temperature is entirely due to the heat generated by the conductive EPDM rubber composite material after energization. The temperature change values at the three locations after multiple tests are the same, indicating that the heat production of the conductive EPDM rubber composite material is uniform. This ensures that the materials produce consistent heat throughout the road, and can achieve comprehensive deicing and snow melting.

### 3.5. Electric Resistance Stability

According to Figure 12 and Table 5, it can be concluded that the conductive EPDM rubber composite material has good electric resistance stability at different input power. During the 30 min of power-on, the electric resistance fluctuates only in a small range. Since the area of the sample for electrothermal test is larger than that of other active snowmelt deicing studies [21,29], the size effect of the composite in practical applications can be effectively reduced. When the input power is 80 W/m^2^, 160 W/m^2^ and 320 W/m^2^, the corresponding average values of the resistance are 0.60 Ω, 0.61 Ω and 0.61 Ω, respectively. The fluctuations are 2.9%, 2.4% and 1.1%, respectively, compared with the average.

Since the sample is directly connected to the alternating current, there is a certain voltage fluctuation. As a result, the corresponding voltage value and the current value have small fluctuations. But the electric resistance is about 0.61 Ω, which has little effect on the heat generation stability of the circuit. According to the results of the resistance stability tests, it can be proven that the conductive EPDM rubber composite material has excellent resistance stability and it can ensure the stable heat generation during use.

### 3.6. Joule Heating Performance of Conductive EPDM Rubber Composite Material

It can be seen from Figure 13 that the joule heating performance of the conductive EPDM rubber composite material is different after inputting different power. The temperature rises faster when the input power is larger. And there is no change at room temperature. As can be seen from Table 6, when the input power is 80 W/m^2^, 160 W/m^2^ and 320 W/m^2^, the corresponding temperature rise rates are 5.6 °C/h, 8.4 °C/h and 12.4 °C/h, respectively. Compared with other active deicing methods [10,37,38], the input power is lower and the temperature rises faster. It shows that the conductive EPDM rubber composite material can more effectively realize road deicing and snow removal. Due to the low input power, the conductive EPDM rubber composite material is more environmentally friendly as an electric heating material, and has a better effect on road deicing and snow melting.

According to Table 6, due to the presence of the thermal insulation layer, the temperature rise rate of the composite surface varies greatly. When the input power is larger, the difference in temperature rise rate between the upper and lower layers is larger. The input power is 80 W/m^2^, 160 W/m^2^ and 320 W/m^2^, and the temperature increase rate is about 3.6 °C/h, 5.2 °C/h and 7 °C/h, respectively. It shows that the thermal insulation layer of conductive EPDM rubber composite material has a good effect. The structural layer design proposed in this study has a good effect. The heat insulation layer can effectively block the downward transfer of heat, so that the heat can be transferred to the upper layer more, and the temperature of the upper layer rises faster. The transfer of heat has a certain directionality, which is beneficial to achieve rapid road deicing and snow melting.

## 4. Conclusions

In this paper, carbon fiber cloth is sandwiched in ethylene propylene diene monomer (EPDM) rubber composite to obtain a conductive EPDM rubber composite material. It has been verified by experiments that the conductive EPDM rubber composite material has excellent durability and heat production efficiency as a road active deicing and snow melting material. This study resulted in the following conclusions:

(1) The addition of the carbon fiber cloth not only makes EPDM rubber have good electrical conductivity, but also effectively improves the mechanical properties of the EPDM rubber. The addition of carbon fiber cloth increases the tensile strength by 78.3% and the tear strength by 23.1%. There is no substantial influence on the compressive strength. The addition of carbon fiber cloth ensures the mechanical properties of conductive EPDM rubber composite material in road use.

(2) Conductive EPDM rubber composite material has excellent high temperature durability. After 72 h high temperature treatment, the tensile strength, tear strength and compressive strength of the conductive EPDM rubber composite material increases by 38.04%, 8.78%, and 33.85%, respectively. After subsequent high temperature treatment, the mechanical properties of the composites tend to be stable. The test results show that the composite material does not exhibit any degradation of mechanical properties at high temperature and has good high temperature durability. This ensures long-term use of the composite material in high temperature environment.

(3) Conductive EPDM rubber composite material has low temperature stability. After 20 cycles of low temperature cycling, the mechanical properties of the composite material show only minor fluctuations and no significant increase or decrease. This proves that the composite material can meet the requirements of long-term use under low temperature circulation.

(4) Through the heat production uniformity tests, a large area of conductive EPDM rubber composite material was electrified, and the temperature variation of the sample was consistent. This proves that the composite material can produce heat uniformly. The feasibility of achieving effective uniform deicing and snow melting is verified.

(5) The composite material developed in this paper has good electrical resistance stability and do not change with heating time and different input power. The material has a low resistance value and achieves rapid temperature rise at low input power. Owing to the presence of the insulation layer, the directional transmission of heat is achieved. This provides an experimental basis for the efficient and environmentally friendly road deicing and snow melting. Subsequent experiments and theoretical studies on the actual deicing and snow melting will be carried out.

In summary, the mechanical and electrothermal tests prove the feasibility and long-term performance of conductive EPDM rubber composite material. At the same time, due to the use of low-cost materials, conductive EPDM rubber composite material can be widely used in snow melting and deicing in road and other fields. The conductive EPDM rubber composite material has broad application prospects.

## Figures and Tables

**Figure 1 polymers-11-02051-f001:**
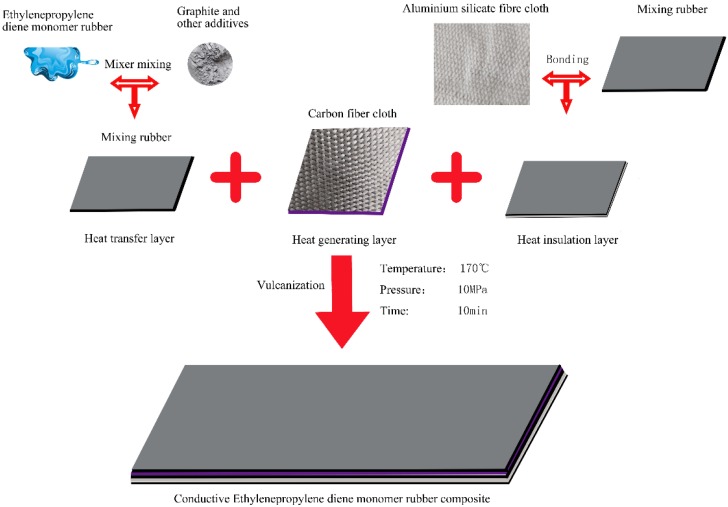
Sample preparation process.

**Figure 2 polymers-11-02051-f002:**
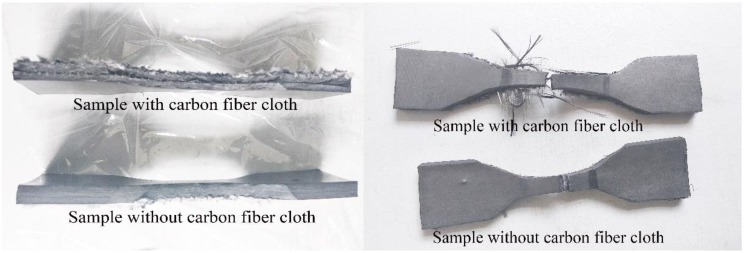
Tensile test specimens and post-test specimens.

**Figure 3 polymers-11-02051-f003:**
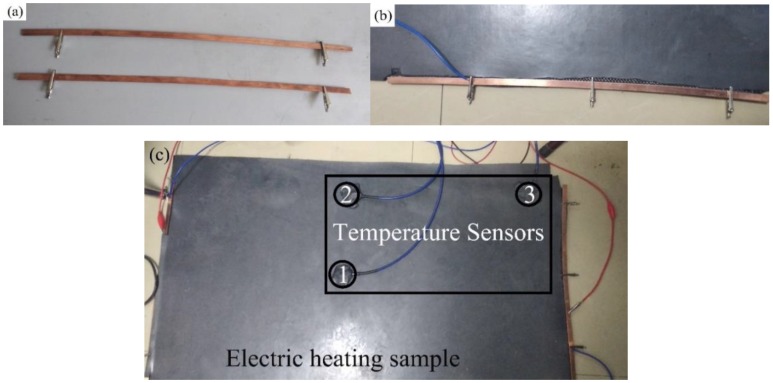
(**a**) Copper electrodes; (**b**) installation position of copper electrode; (**c**) 1,2 and 3 are temperature sensors and their distribution positions.

**Figure 4 polymers-11-02051-f004:**
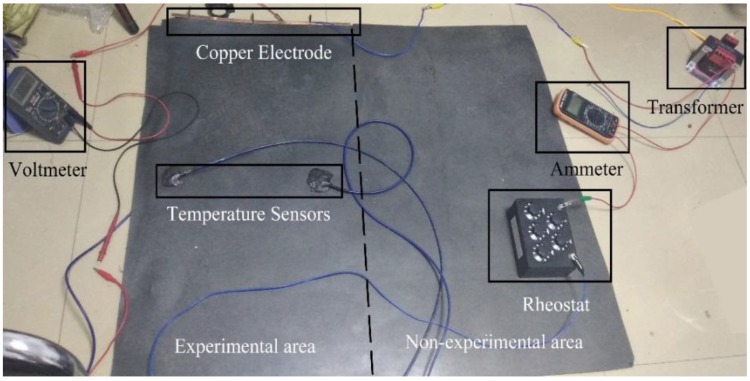
Circuit connection and electrical components.

**Figure 5 polymers-11-02051-f005:**
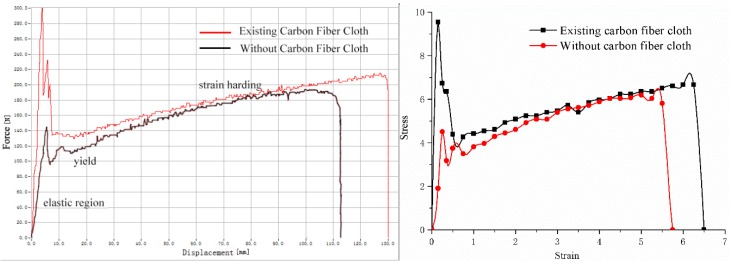
Force-displacement curves (**left**) and Stress-strain curves of tensile tests (**right**).

**Figure 6 polymers-11-02051-f006:**
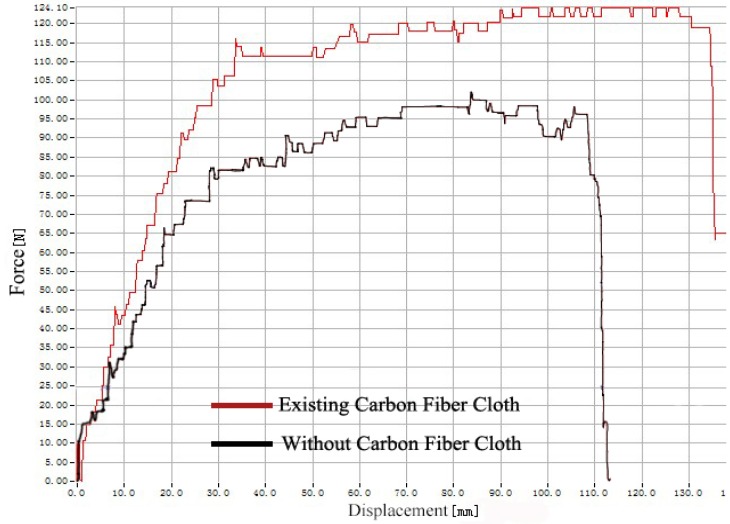
Force-displacement curves of tear tests.

**Figure 7 polymers-11-02051-f007:**
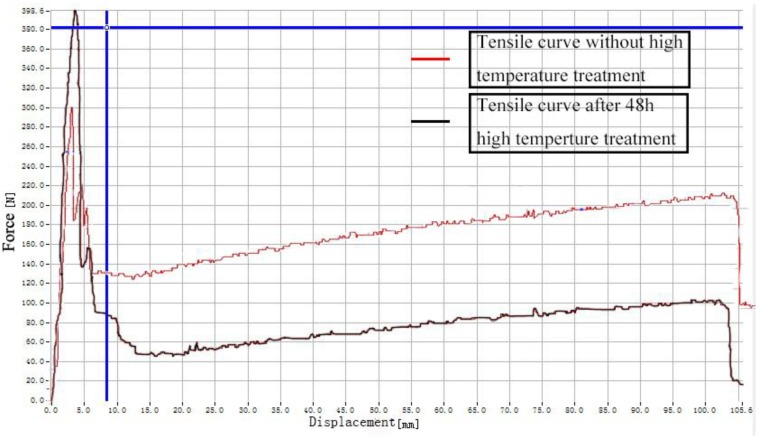
Effect of high temperature on tensile curves of conductive ethylene propylene diene monomer (EPDM) rubber composite.

**Figure 8 polymers-11-02051-f008:**
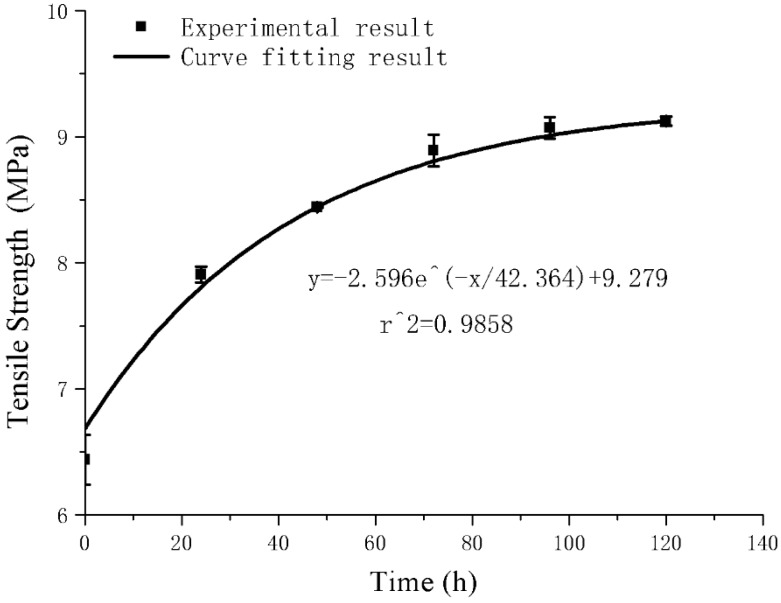
Effect of high temperature duration on tensile strength of conductive EPDM rubber composite and curve fitting.

**Figure 9 polymers-11-02051-f009:**
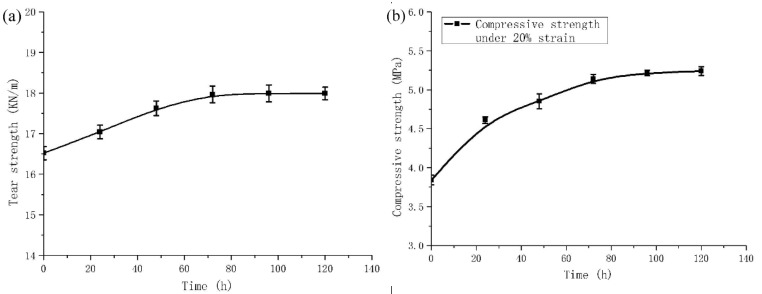
The effect of high temperature on (**a**) tear strength; (**b**) compressive strength.

**Figure 10 polymers-11-02051-f010:**
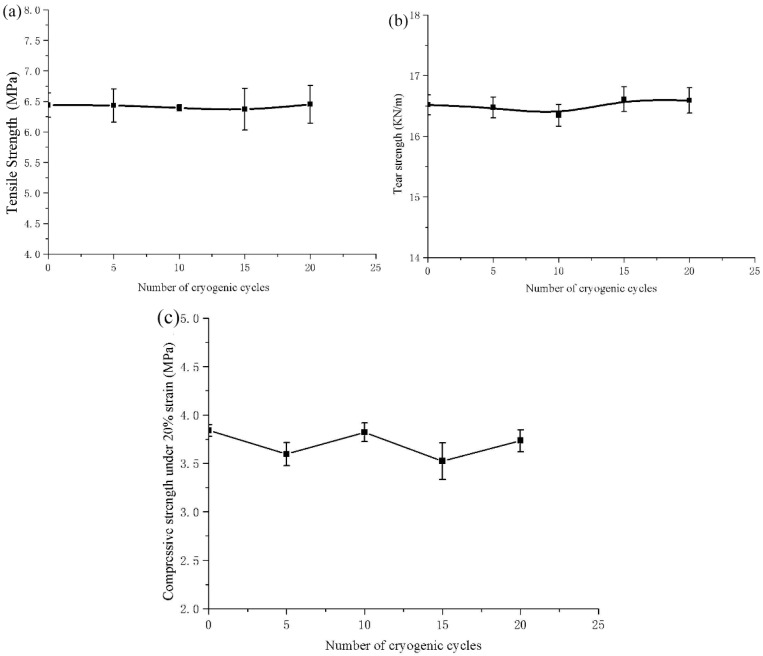
Effect of low temperature cycle times on (**a**) tensile strength; (**b**) tear strength; (**c**) compressive strength of conductive EPDM rubber composite.

**Figure 11 polymers-11-02051-f011:**
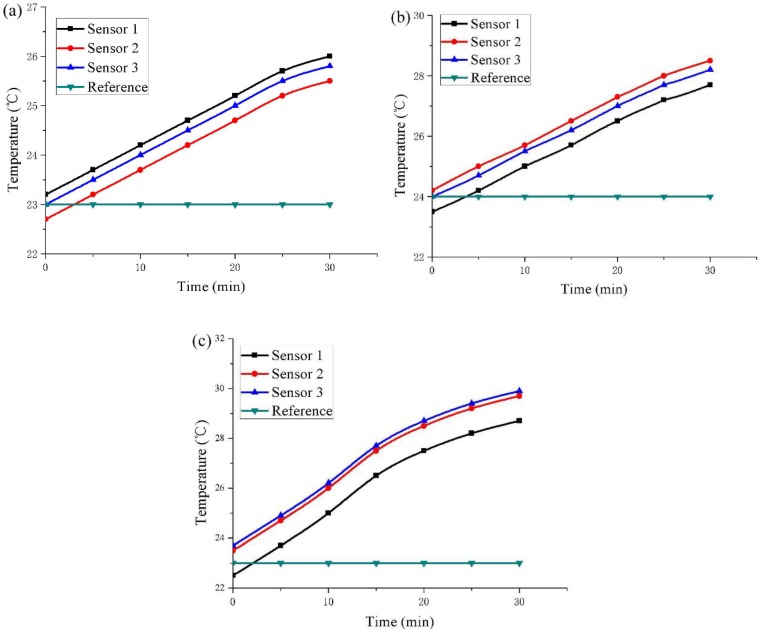
Different input power (**a**) 80 W/m^2^; (**b**) 160 W/m^2^; (**c**)320 W/m^2^, the temperature of the three positions with the power-on time.

**Figure 12 polymers-11-02051-f012:**
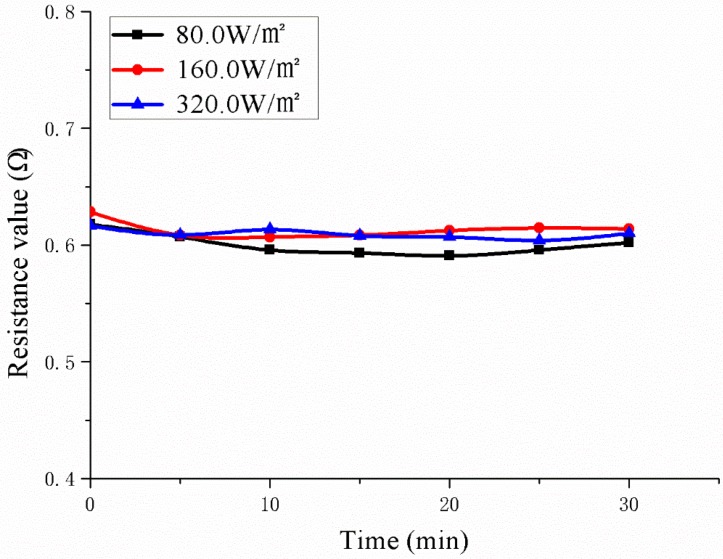
The variation of resistance with power-on time at different input power.

**Figure 13 polymers-11-02051-f013:**
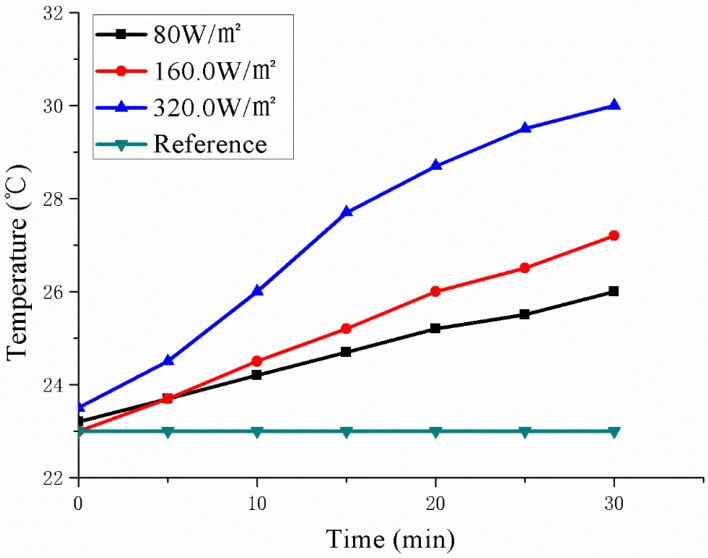
Temperature changes with energization time under different input power.

**Table 1 polymers-11-02051-t001:** Effect of carbon fiber cloth on mechanical properties of composite material.

		Tensile Strength/(MPa)			Tear Strength/(KN/m)			Compressive Strength/(MPa)		
**Without Carbon Fiber Cloth**	**Experiment**	3.70	3.52	3.63	13.43	13.57	13.26	3.87	3.84	3.84
	**Average ± Standard Deviation**	3.61 ± 0.075			13.42 ± 0.127			3.85 ± 0.014		
**Existing Carbon Fiber Cloth**	**Experiment**	6.30	6.58	6.44	17.04	16.61	15.91	3.88	3.85	3.89
	**Average ± Standard Deviation**	6.44 ± 0.114			16.52 ± 0.466			3.87 ± 0.017		

**Table 2 polymers-11-02051-t002:** Effect of high temperature heating time on mechanical properties of material.

**Heating Time/h**	0	24	48	72	96	120
**Tensile Strength/(MPa)**	6.44	7.91	8.44	8.89	9.07	9.13
**Tear Strength/(KN/m)**	16.52	17.04	17.62	17.97	17.99	17.99
**Compressive Strength/(MPa)**	3.84	4.61	4.85	5.14	5.22	5.24

**Table 3 polymers-11-02051-t003:** Effect of low temperature cycling on mechanical properties of the material.

**Number of Cryogenic Cycles**	0	5	10	15	20
**Tensile Strength/(MPa)**	6.44	6.43	6.39	6.37	6.45
**Tear Strength/(KN/m)**	16.52	16.48	16.35	16.61	16.59
**Compressive Strength/(MPa)**	3.84	3.60	3.82	3.52	3.73

**Table 4 polymers-11-02051-t004:** Upper layer temperature of the sample after 30 min of energization.

**Input Power/(W/m^2^)**	80			160			320		
**Temperature Sensor**	1	2	3	1	2	3	1	2	3
**Initial Temperature/°C**	23.2	22.7	23.0	23.5	24.2	24.0	22.5	23.5	23.7
**Power on for 30 min/°C**	26.0	25.5	25.8	27.7	28.4	28.2	28.7	29.7	29.9
**Difference Value/°C**	2.8	2.8	2.8	4.2	4.2	4.2	6.2	6.2	6.2

**Table 5 polymers-11-02051-t005:** Resistance change with energization time at different input power.

**Time/min**		0	5	10	15	20	25	30
**Resistance Value/Ω**	Input 80W/m^2^	0.618	0.607	0.596	0.593	0.591	0.596	0.602
	Input 160W/m^2^	0.628	0.609	0.607	0.609	0.612	0.615	0.614
	Input 320W/m^2^	0.617	0.609	0.613	0.608	0.607	0.604	0.610

**Table 6 polymers-11-02051-t006:** Temperature rise rate under different input power.

**Heat Flux Density/(W/m^2^)**		80	160	320
**Temperature Rise Rate/(°C/h)**	Upper layer	5.6	8.4	12.4
	Lower layer	2.0	3.2	5.4

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
