# Peer review of "Durability and Electrical Conductivity of Carbon Fiber Cloth/Ethylene Propylene Diene Monomer Rubber Composite for Active Deicing and Snow Melting"

_polymers, 2019, doi:10.3390/polym11122051_

Round 1

Reviewer 1 Report

Line #  Comment / Question

137     Was wear of the material considered?

359     Was cyclic “fatigue” of the material considered, i.e., many cycles of heating / cooling?

Reviewer 2 Report

In this work the durability and the electrical conductivity of a carbon fiber cloth/ethylene propylene diene monomer rubber composite is studied to evaluate the possibility to use this composite material for active deicing and snow melting.

The motivation of the study is well explaned and the work is interesting. The experimental part is clear and the data and analyses are presented appropriately.

In my opinion the paper can be accepted for publication after minor revisions.

Comments:

Line 46: please explain better the sentence: “In addition, the electric resistance value of the mixture is larger and the thermal efficiency is lower”. To which mixture are you referring to? The electric resistance is  larger compared to what?

Line 64: please use comma instead of stop mark.

Line 76: please use comma instead of stop mark.

Line 135: please use comma instead of stop mark.

Line 142: please add the geometric characteristics of the tensile specimens that have been tested.

Line 154: the image of the mechanical specimens before tests is not very clear: please change it to better appreciate the differences between the two materials. Please label the two samples showed in the image of the tested samples.

Line 160: please replace the comparative “greater” with “great”.

Line 197: please replace “was” with “were”.

Line 200: please show also a stress-strain curve for the tensile test.

Line 204: please add the standard deviation to the average values and improve consistency of column alignment in the table.

Line 210: please check the use of italic characters in the formula (1).

Line 226: the sentence “but the compressive strength of the composite is only 0.5%” is not clear, please modify it.

Line 321: in the table please correct “valuer/W”
